# What Do Learning Dynamics Reveal About Generalization in LLM Mathematical Reasoning?

**Katie Kang** [1]   **Amrith Setlur** [2]   **Dibya Ghosh** [1]
**Jacob Steinhardt** [1]   **Claire Tomlin** [1]   **Sergey Levine** [1]   **Aviral Kumar** [2]

## Abstract

Modern large language models (LLMs) excel at fitting finetuning data, but often struggle on unseen examples. In order to teach models genuine reasoning abilities rather than superficial pattern matching, our work aims to better understand how the learning dynamics of LLM finetuning shapes downstream generalization. Our analysis focuses on reasoning tasks, whose problem structure allows us to distinguish between memorization (the exact replication of reasoning steps from the training data) and performance (the correctness of the final solution). We find that a model's performance on test prompts can be effectively characterized by a training metric we call *pre-memorization train accuracy*: the accuracy of model samples on training queries before they begin to copy the exact reasoning steps from the training set. On the dataset level, this metric is able to almost perfectly predict test accuracy, achieving $R^2$ of $\geq 0.9$ across various models (Llama3 8B, Gemma2 9B), datasets (GSM8k, MATH), and training configurations. On a per-example level, this metric is also indicative of whether individual model predictions are robust to perturbations in the training query. By connecting a model's learning dynamics to test performance, pre-memorization train accuracy can inform training decisions, such as the makeup of the training data. Our experiments on data curation show that prioritizing examples with low pre-memorization accuracy leads to 1.5-2x improvements in data efficiency compared to i.i.d. data scaling and other data scaling techniques.

---

*Equal contribution  [1]UC Berkeley [2]CMU. Correspondence to: Katie Kang <katiekang@eecs.berkeley.edu>.

*Proceedings of the 42$^{nd}$ International Conference on Machine Learning*, Vancouver, Canada. PMLR 267, 2025. Copyright 2025 by the author(s).

## 1. Introduction

Large language models (LLMs) have demonstrated remarkable problem-solving capabilities, yet the mechanisms by which they learn and generalize remain largely opaque. For instance, consider a set of LLMs, each derived from the same pretrained model and finetuned on the same reasoning dataset but with varying learning rates (Fig. 1). While several of these models reach near-perfect accuracy on training data, their test performances were vastly different. This raises the question: *what factors in an LLM's finetuning training lead to differences in its generalization behavior?* Understanding these factors could help us design better training methods that foster genuine reasoning abilities in models, rather than mere pattern matching.

We focus on mathematical problem-solving tasks, whose structure is particularly amenable for investigating this question. In reasoning tasks, models are trained to generate both a final answer and intermediate reasoning steps. Although each problem has a single correct answer, the reasoning steps in the target solution trace represent just one of many valid ways to solve a problem. Therefore, a model that has memorized the training data is likely to replicate exact reasoning steps from the training data, while a model with general problem-solving skills may produce the correct final answer but follow a different reasoning path. By analyzing model responses on training queries, focusing on both the accuracy of the final answer and the similarity of the response to the target solution trace, we can gain insights into the generalizability of the model's learned solution.

Our findings reveal that, while LLMs often fully memorize the finetuning dataset by the end of training, model predictions for training queries *prior to memorization* are strongly indicative of final test performance. For certain examples, models first learn to generate diverse solution traces (distinct from the target solution trace) that lead to the correct final answer, before later memorizing the target solution trace. For other training examples, models only produce incorrect responses before memorizing the target trace. To capture this distinction, we introduce the concept of *pre-memorization train accuracy*: the highest accuracy a model achieves on a training example through the course

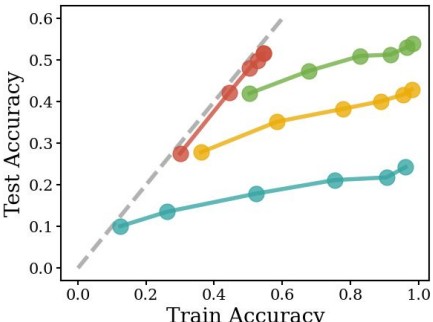 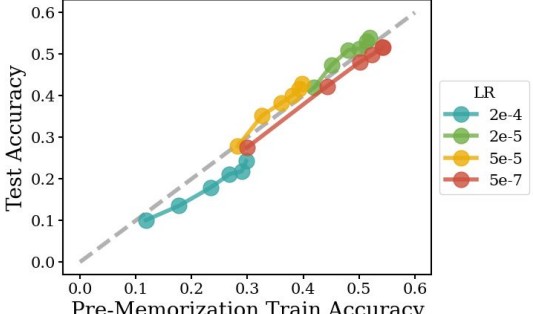

*Figure 1.* Relationship between train accuracy (left), pre-memorization train accuracy (right), and test accuracy for models finetuned on GSM8k using Llama3 8B. Each line represents a training run, and each point represents an intermediate checkpoint. Pre-memorization train accuracy strongly correlates with test accuracy, while train accuracy does not.

of training before exactly memorizing the target solution trace. We find that a model's average pre-memorization train accuracy is highly predictive of the model's test accuracy, as illustrated in Fig. 1. Our experiments show that this phenomenon holds across different models (e.g., Llama3 8B (Dubey et al., 2024), Gemma2 9B (Team et al., 2024)), tasks (e.g., GSM8k (Cobbe et al., 2021), MATH (Hendrycks et al., 2021)), dataset sizes, and hyperparameter settings, with coefficients of determination around or exceeding 0.9.

We further find that the pre-memorization train accuracy can provide insights into the robustness of model predictions at a per-example level. For train examples with low pre-memorization accuracies, adding small perturbations to the training prompt causes the accuracy of model predictions to significantly degrade. In contrast, for train examples with high pre-memorization accuracies, models are generally able to maintain high performance under perturbations. Thus, by measuring pre-memorization accuracy, we can identify specific training examples for which a model's predictions are not robust, which can inform targeted improvements to the training strategy. As an example, we leverage our findings to guide data curation. Our experiments show that training on data distributions that prioritize examples with low pre-memorization accuracy leads to a 1.5-2× improvement in sample efficiency over i.i.d sampling, and outperforms other standard curation techniques.

The main contributions of this work are as follows: **(1)** we introduce the concept of pre-memorization train accuracy, and show that it is highly predictive of test accuracy for LLM reasoning problems, **(2)** we show that pre-memorization train accuracy can also predict the robustness of individual model predictions for train examples, and **(3)** we leverage our observations to improve the sample efficiency of data curation. By offering a deeper understanding of how a model's learning dynamics shape its generalization, we hope our work can bring about more targeted and principled interventions for improving a model's reasoning capabilities.

## 2. Related Works

A number of works have studied the phenomenon of memorization during training, but consider different definitions of memorization. One definition quantifies memorization with the "leave-one-out" gap, i.e., how much a model's prediction for an example changes if we were to remove it from the training data (Feldman & Zhang, 2020; Arpit et al., 2017; Zhang et al., 2017). Using this definition, some works argue that more memorization during training leads to worse generalization (Bousquet & Elisseeff, 2000), while others contend that memorization is actually necessary for generalization in long-tail distributions (Feldman, 2020). These works generally produce worst-case bounds on generalization error within some class of training distributions. In contrast, our work presents a direct, empirical connection between a model's train behavior and its test accuracy without relying on the computationally expensive "leave-one-out" metric. In the context of language models, others have defined memorized examples as those where the model's output closely matches examples in the training data (Carlini et al., 2021; Tirumala et al., 2022; Inan et al., 2021; Hans et al., 2024), which is similar to our definition of memorization. However, these works mainly focus on privacy and copyright concerns, rather than connections to generalization.

Beyond memorization, a number of prior works have studied how other aspects of the learning process relate to generalization. Some works focus on metrics related to model complexity, such as VC dimension or parameter norms (Neyshabur et al., 2015; Bartlett et al., 2019), while other works focus on empirically motivated measures, such as gradient noise (Jiang et al., 2019) or distance of trained weights from initialization (Nagarajan & Kolter, 2019). Jiang et al. (2019) conducted a comprehensive comparison of these measures and found that none were consistently predictive of generalization, though their work primarily focused on image classification. Other approaches have used unlabeled, held-out data to predict generalization, leverag-

ing metrics such as the entropy of model predictions or the disagreement between different training runs (Garg et al., 2022; Platanios et al., 2016; Jiang et al., 2021). Our findings show that pre-memorization accuracy can be a much stronger predictor of generalization in LLM reasoning tasks.

Finally, our work seeks to improve data curation, which has also been studied in a number of prior works. Specific to LLM finetuning, prior data curation approaches largely fall into three categories: optimization-based, model-based, and heuristic-based approaches. Optimization-based methods frame data selection as an optimization problem, where the objective is model performance, and the search space consists of the training data distribution (Engstrom et al., 2024; Grosse et al., 2023). Model-based approaches leverage characteristics of the learning process (Mekala et al., 2024; Liu et al., 2024), such as comparing the perplexity of examples (Li et al., 2023). Lastly, heuristic-based methods rely on simpler criteria, such as difficulty scores generated by off-the-shelf LLMs such as GPT, to classify desirable training data (Chen et al., 2023; Lu et al., 2023; Zhao et al., 2023). Our data curation approach aligns most closely with model-based strategies, as we use the model's pre-memorization accuracy, a characteristic of the learning process, to inform the selection of training examples. Our experiments show that pre-memorization train accuracy can serve as an effective metric for data curation, and outperforms previous approaches.

## 3. Preliminaries

We focus on training LLMs to perform reasoning tasks via finetuning. We are provided with a training dataset $D_{\text{train}} = \{(x_i, y_i)\}$, where queries $x_i$ are drawn from $P(x)$ and solution traces $y_i$ are drawn from $P(y|x)$. We assume the test dataset, $D_{\text{test}}$, is generated from the same distribution as the training data. The model is finetuned by minimizing next-token prediction loss. We denote the finetuned model as $f_\theta(y|x)$, and model predictions as $\hat{y} \sim f_\theta(y|x)$.

In reasoning tasks, solution traces $y$ consist of both intermediate reasoning steps and a final answer, denoted as $\textbf{Ans}(y)$. Our goal in a reasoning task is for the model to generate solution traces with the correct final answer when faced with previously unseen queries. We measure the accuracy of model samples for a given query $x_i$ using $\textbf{Acc}(f_\theta(y|x_i), y_i) = \mathbb{E}_{\hat{y}_i \sim f_\theta(y|x)}[\mathbb{1}(\textbf{Ans}(\hat{y}_i) = \textbf{Ans}(y_i))]$. In our experiments, we approximate this accuracy by sampling from the model with a temperature of 0.8 and averaging the correctness attained by the samples.

While different solution traces drawn from $P(y|x)$ should all have the same final answer, the target solution trace $y_i$ of an example represents only one of many valid solution traces for solving $x_i$. Thus, model samples for a

train query may contain reasoning steps that differ from the target solution trace, while still arriving at the correct final answer. To quantify this difference, we will measure the distance between a model's prediction $f_\theta(y|x_i)$ and the target reasoning trace $y_i$ with perplexity, defined as $\textbf{Perp}(f_\theta(y|x_i), y_i) = \exp(\frac{-1}{n_i} \log(f_\theta(y_i|x_i)))$, where $n_i$ is the number of tokens in $y_i$.

## 4. Connecting Learning Dynamics to Generalization

In this section, we will investigate the relationship between a model's learning dynamics during finetuning and its ability to generalize. Our findings show that, while models tend to memorize most of the training data after some number of epochs, their generated samples display varying levels of accuracy before memorization occurs. We find that this accuracy before memorization has a strong connection to the model's downstream generalization behavior.

### 4.1. Characterizing the Learning Dynamics of LLM Reasoning Finetuning

We begin by more precisely characterizing an LLM's learning process when finetuning on reasoning tasks. We focus on two key aspects of the model's behavior when presented with train queries: (1) whether the model's samples arrive at the correct final answer, and (2) the distance between the model's prediction and the target solution trace, measured by perplexity. These two metrics, visualized in Fig. 2, offer different perspectives on the model's behavior, because while there is only one correct final answer for each query, there may exist many different valid reasoning traces. Tracking both metrics through the course of training allows us to measure how effectively the model is able to solve training queries, and the extent to which this is accomplish by replicating the target solution trace.

In Fig. 3, we visualize the learning progression, as characterized by the two metric described above, for three models finetuned on GSM8K. Each model is trained for six epochs, with a distinct peak learning rate that decays to zero by the end of training. As expected, training accuracy improves over time as the model minimizes the loss (color gradient from dark to light), and the distance between predictions and target solution traces decrease (from pink to yellow). For some learning rate settings, models approach near-perfect accuracy by the end of training, and their predictions closely match the target reasoning traces (mostly yellow in bottom row). However, during early stages of training, we observe significant differences in model behavior. For some train queries, models initially produce incorrect samples (black), and then directly transition to replicating the target trace (yellow). For other examples, models first learn to generate correct answers with solution traces that differ from the tar-

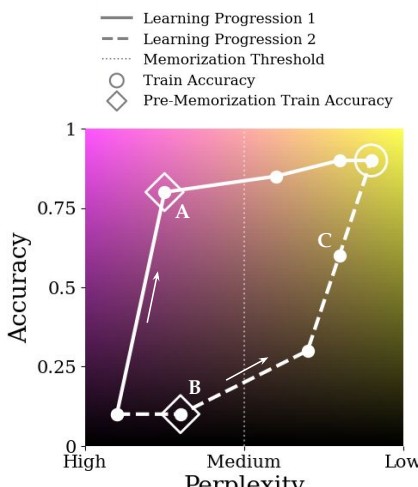

**Train Example**

**Prompt:** Natalia sold clips to 48 of her friends in April, and then she sold half as many clips in May. How many clips did Natalia sell altogether in April and May?

**Solution Trace:** Natalia sold 48/2 = <<48/2=24>>24 clips in May. Natalia sold 48+24 = <<48+24=72>>72 clips altogether in April and May. #### **72**

**Model Samples**

**A:** In May, Natalia sold half as many clips as in April, so she sold 48 / 2 = <<48/2=24>>24 clips. She sold altogether 48 + 24 = <<48+24=72>>72 clips in April and May. #### **72**

**B:** In April, Natalia sold 48 * 2 = <<48*2=96>>96 clips. In May, Natalia sold 96 / 2 = <<96/2=48>>48 clips. Natalia sold 96 + 48 = <<96+48=144>>144 clips altogether in April and May. #### **144**

**C:** Natalia sold 48/2 = <<48/2=24>>24 clips in May. Natalia sold 48+24 = <<48+24=72>>72 clips altogether in April and May. #### **72**

*Figure 2.* Visualizations of different learning progressions, as measured by the accuracy of model samples (light vs. dark) and the perplexity of target solution traces under model predictions (pink vs. yellow). Right side presents examples of model samples with (A) high accuracy+high perplexity, (B) low accuracy+high perplexity, and (C) high accuracy+low perplexity. Black text represents exact match with the target solution trace, while grey text represents parts that do not match.

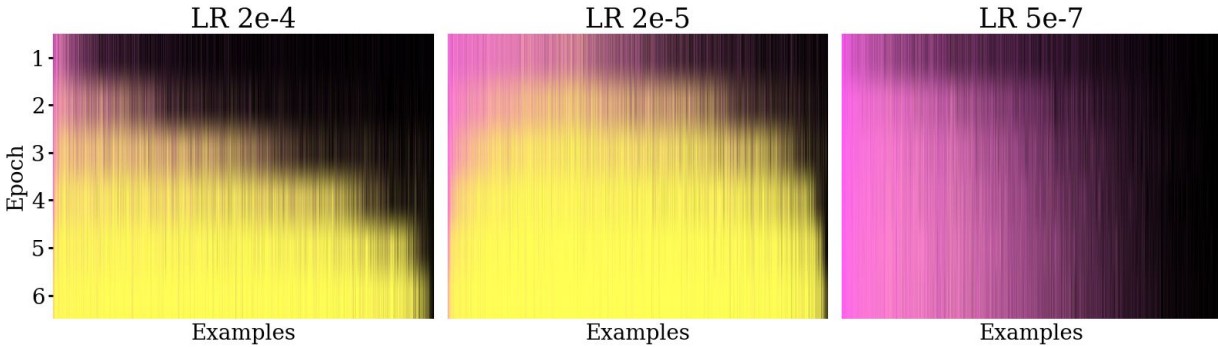

*Figure 3.* Predictions of 3 different models through the course of training. X-axis represents individual training examples. Y-axis represents the training epoch. Color represents model predictions for each example in terms of accuracy and perplexity (legend in Fig. 2).

get trace (pink), before later transitioning to fully replicating the target trace (yellow).

In this work, we will refer to model predictions with low distance to target solution traces as *memorization*. We can see that when finetuned with different learning rates, different models exhibit different capacities for generating accurate samples before memorizing target solution traces (amount of pink). Models with low accuracy before memorization may be largely learning verbatim mappings from training queries to target traces, which would not generalize to new queries. In contrast, models with high accuracy before memorization demonstrate an ability to arrive at correct answers through varied reasoning paths, suggesting that they have developed more generalizable problem-solving capabilities.

To better quantify this phenomenon, we introduce a metric called pre-memorization accuracy. We consider a train example $(x_i, y_i) \in D_{\text{train}}$ to be memorized by $f_\theta(y|x)$ if

$\mathbf{Perp}(f_\theta(y|x_i), y_i) < p$, where $p$ is a threshold (fixed across examples). We further define a modified measure of accuracy, whose value is masked to zero if the model's prediction for that example is considered memorized, as follows:

$$\mathbf{MaskedAcc}(f_\theta(y|x_i), y_i, p)$$
$$=\mathbf{Acc}(f_\theta(y|x_i), y_i) \cdot \mathbb{1}[\mathbf{Perp}(f_\theta(y|x_i), y_i) > p].$$

Now let $f_{\theta_m}$ denote the model at epoch $m$ of training. Using our definition of masked accuracy, we define the **pre-memorization accuracy** as follows:

$$\mathbf{PreMemAcc}(f_{\theta_{1:m}}(y|x_i), y_i, p)$$
$$=\min\Big\{\max_{1\leq m'\leq m} \mathbf{MaskedAcc}(f_{\theta_{m'}}(y|x_i), y_i, p),$$
$$\mathbf{Acc}(f_{\theta_m}(y|x_i), y_i)\Big\}$$

This quantity can be roughly interpreted as the best accuracy that the model achieves for a training prompt thus far

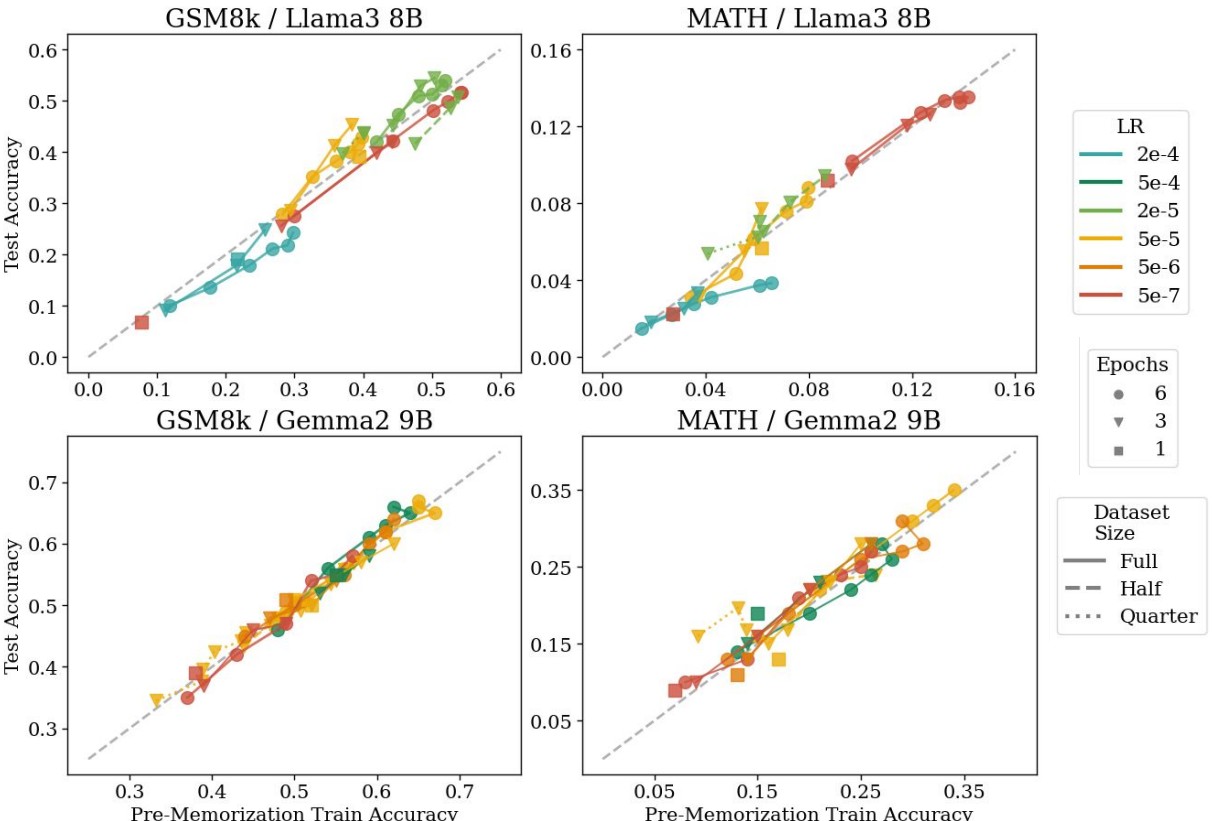

*Figure 4.* Evaluating the relationship between pre-memorization train accuracy and test accuracy. Each line corresponds to a training run, and each marker corresponds to a specific checkpoint. Pre-memorization train accuracy strongly predict test accuracy across tasks, models, and training settings.

in training before it memorizes the target trace. Unlike standard accuracy, which evaluates performance at specific checkpoints, pre-memorization accuracy evaluates the entire training process up to epoch $m$. There is an additional minimum taken with the accuracy of model predictions at epoch $m$, which compensates for examples whose accuracies decrease through training (though this is uncommon).

### 4.2. Pre-Memorization Train Accuracy Strongly Predicts Test Accuracy

We next use pre-memorization accuracy to analyze the connection between learning dynamics and downstream generalization. We find that **a model's average pre-memorization train accuracy is highly predictive of its test accuracy** across a variety of training runs and checkpoints. More concretely, we find that there exists a value of $p$ for which a model's average pre-memorization train accuracy, $\mathbb{E}_{D_{\text{train}}}[\textbf{PreMemAcc}(f_{\theta_{1:m}}(y|x_i), y_i, p)]$, closely approximates the model's test accuracy, $\mathbb{E}_{D_{\text{test}}}[\textbf{Acc}(f_{\theta_m}(y|x_i), y_i)]$. The value of the memorization threshold $p$ is fixed across examples and training parameters, but may need to be recalibrated for different tasks or models. We calibrate $p$ by

sweeping across a range of values (see Appendix A).

In Fig. 4, we plot the pre-memorization training accuracy and test accuracy across different training runs. We used Llama3 8B and Gemma2 9B as base models and GSM8K and MATH as the reasoning tasks. To evaluate different generalization behaviors, we finetuned the models by adjusting the peak learning rate (ranging from 5e-7 to 5e-4), the number of training epochs (1, 3, 6), and the dataset size (full, half, or quarter of the original dataset). We use the same value for $p$ within each plot. A full list of the training runs in our experiments and other details can be found in Appendix B. We observe a strong linear relationship between pre-memorization training accuracy and test accuracy, with the results closely following the $y = x$ line across different models, tasks, and hyperparameter settings. More quantitatively, the coefficients of determination associated with each plot are 0.94 (GSM8k Llama), 0.95 (MATH Llama), 0.97 (GSM8k Gemma), and 0.88 (MATH Gemma). Our results show that pre-memorization training accuracy is a reliable predictor of test accuracy.

As discussed in Section 2, various metrics have been proposed in previous studies to predict the generalization gap,

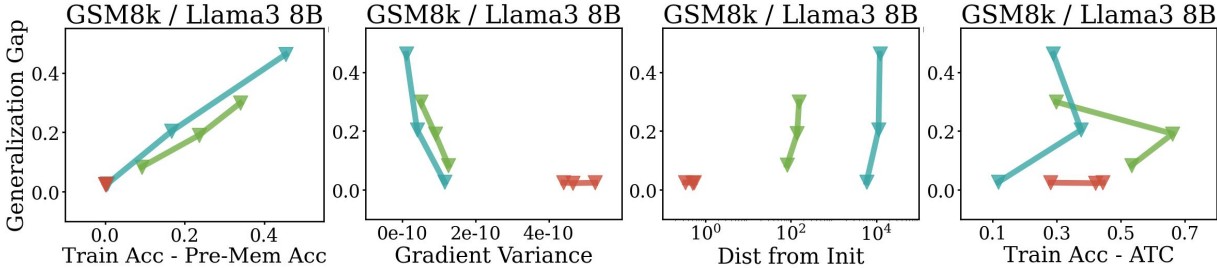

*Figure 5.* Evaluating different generalization metrics vs. the ground truth generalization gap for models finetuned on GSM8k using Llama3 8B (legend in Fig. 4).

the difference between train and test accuracy. In Fig. 5, we compare several of these existing metrics, including gradient variance (Jiang et al., 2019), distance between current model weights and initialization (Nagarajan & Kolter, 2019), and an estimate of test accuracy via Average Thresholded Confidence (ATC) (Garg et al., 2022) (details in Appendix C). The correlation coefficients associated with each metric (left to right) are 0.98, -0.72, 0.59, -0.04, which shows that the prior metrics do not correlate as strongly with test accuracy as our proposed metric.

## 5. Per-Example Analysis of Generalization

In this section, we go beyond aggregate test accuracy and show that tracking per-example pre-memorization accuracy offers a window into the model's behavior at the level of individual training examples. Specifically, we find that the pre-memorization train accuracy of a given example is predictive of the robustness of the model's prediction for that example. This example-level accuracy helps us identify subsets of the training data for which the model struggles to learn robust solutions and offers opportunities to improve training through targeted interventions. We explore how this insight can inform data curation strategies, showing that prioritizing examples with low pre-memorization train accuracy during data collection can lead to significant improvements over i.i.d. data collection and other common data curation methods.

### 5.1. Predicting Model Robustness with Pre-Memorization Train Accuracy

We begin by examining the relationship between an individual example's pre-memorization train accuracy and the robustness of the model's predictions for that example. Our findings show that **model predictions tend to be less robust for train examples with low pre-memorization accuracy**.

To assess the robustness of model predictions, we analyze how the model responds to small perturbations in the input prompt. We present the model with both the original training queries, as well as training queries appended with short

preambles to the solution trace—phrases such as "First" or "We know that"—that deviate from the target solution trace, which we visualize in Fig. 6. Because these generic phrases are plausible preambles to valid reasoning traces, we would expect a model which has learned a robust solution to an example to still be able to arrive at the correct final answer. In contrast, if the model is unable to produce the correct final answer given these generic phrases, then the model is likely to have learned to only regurgitate the training response.

In Fig. 6, we show the prediction behavior of two models, both trained for six epochs with a learning rate of 2e-5, on the GSM8K and MATH datasets. We can see that while model predictions are near-perfect for unaltered training prompts, their accuracy significantly degrades when presented with perturbed prompts. Furthermore, we see that the accuracy of train examples with low pre-memorization train accuracy tends to degrade much more than those with high pre-memorization train accuracy. These findings suggest that pre-memorization train accuracy can predict the robustness of model predictions for individual train examples. Note that while our perturbation analysis makes use of manually-constructed, task-dependent preambles, pre-memorization train accuracy does not require any domain knowledge. Therefore, pre-memorization train accuracy provides a practical way to identify fragile examples where the model may have learned overly specific or non-robust patterns, which offers practical applications for improving model generalization.

### 5.2. Curating Data with Pre-Memorization Train Accuracy

By offering insights into the robustness of individual model predictions, pre-memorization train accuracy can provide targeted guidance for improving a model's generalization. In this section, we explore data curation as a practical application of our findings. Prior work has suggested that focusing on "harder" examples, where the model struggles to learn robust solutions, can lead to more sample-efficient improvements (Li et al., 2023; Chen et al., 2023). However, identifying useful metrics for determining example diffi-

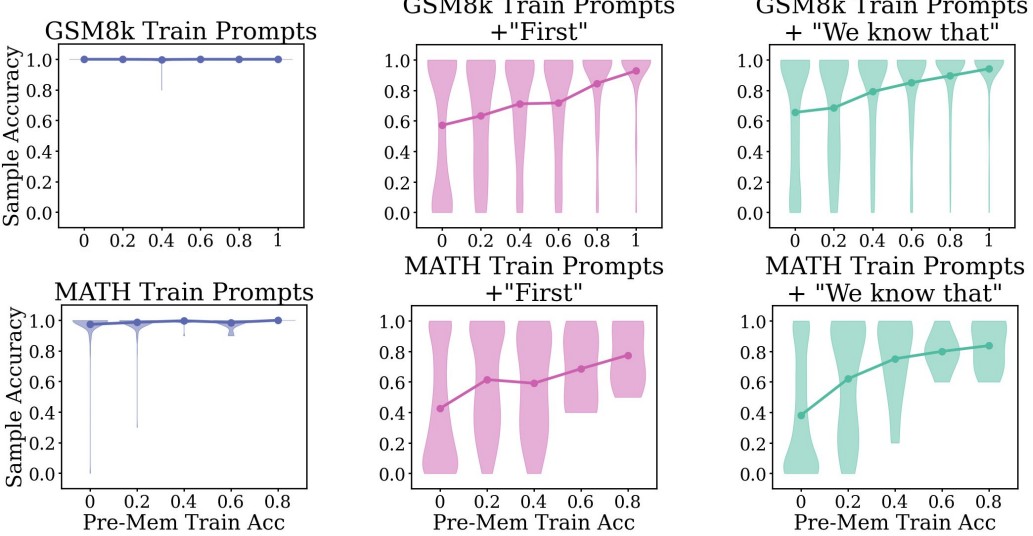

**Robust Solution**

**Non-Robust Solution**

*Figure 6.* Visualization of the robustness of model predictions to perturbations in the prompt, including the original training prompt (purple), original prompt + "First" (pink), and original prompt + "We know that" (teal). A robust model prediction would arrive at the correct final answer even if the perturbations changes the reasoning steps. In contrast, a non-robust model prediction produces incorrect final answer when the prompt diverges from the training data.

*Figure 7.* Accuracies of model samples (y-axis) when faced with the original prompt (left) and prompts with perturbations (middle, right). The x-axis represents bins of pre-memorization train accuracies associated with each prompt. Solid line denotes the average, and violins denote distributions within each bin. While the accuracy of model samples is almost perfect when faced with original prompts, it significantly degrades when faced with prompts with perturbations. Furthermore, the degradation of accuracy is much more significant for train examples with low pre-memorization accuracy than those with high pre-memorization accuracy, showing that per-example pre-memorization train accuracy can provide insight into the robustness of a model's individual predictions.

culty remains an open challenge. We investigate the use of pre-memorization train accuracy as a metric for guiding data curation, and find that it outperforms i.i.d. sampling and other standard data curation approach in sample efficiency for reasoning tasks.

We will first more precisely define our data curation problem. Given an existing set of $N$ training examples with queries distributed as $P(x)$, we aim to collect $N'$ examples, denoted as $D'_{\text{train}}$, to augment the dataset. The goal is to specify a new distribution $P'(x)$ that maximizes the test performance of a model trained on both the original and the newly collected examples. While defining the true distribution of queries can be challenging, we assume that by approximating it with an empirical distribution from the current dataset, we can collect new data with similar prop-

erties. In our experiments, we take $D_{\text{train}}$ to be the original dataset, and collect new examples by using GPT to rephrase examples in the original dataset, similar to the procedure in (Setlur et al., 2024). By only collecting new examples that derive from the specified empirical distribution, we can ensure the new dataset approximates $P'(x)$. This setup can also be used when collecting new human-generated data, by providing the specified empirical distribution of examples as references for human labelers.

Our approach for data collection prioritizes examples with low pre-memorization accuracy. First, we calculate the pre-memorization accuracy for each example in the current dataset and then define $P'(x)$ as the distribution of examples whose pre-memorization accuracy falls below a certain threshold $t$. We then collect new data according to this dis-

**Algorithm 1** Our Data Collection Process

1: **Input:** $N' = N'_1 + \cdots + N'_n, t$
2: **Output:** Updated dataset $D'_{\text{train}}$
3: Initialize $D'_{\text{train}} = \{\}$
4: **for** $i = 1$ to $n$ **do**
5:     Train model on $D_{\text{train}} + D'_{\text{train}}$
6:     Evaluate model on $D_{\text{train}}$ and compute pre-memorization accuracy for each example
7:     Set $P'_i(x)$ as the distribution of examples with pre-memorization accuracy below $t$
8:     Collect $N'_i$ new examples from $P'_i(x)$ and add them to $D'_{\text{train}}$
9: **end for**

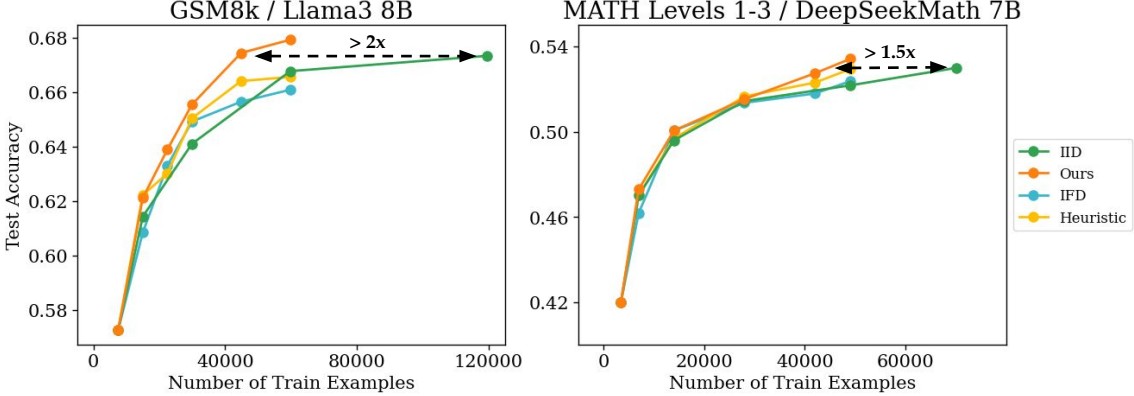

*Figure 8.* Comparison of different approaches for data curation. Each line represents a different data curation approach, and each point represents a different training run. Our approach acheived the best sample efficiency compared to the other approaches.

tribution. If $N'$ is large, we can split the data collection process into multiple iterations ($N'_1 + ... + N'_n = N'$). In each iteration, we collect $N'_i$ new examples according to $P'_i(x)$, retrain a model on the combined dataset, calculate the pre-memorization accuracy with the model, and update $P'_{i+1}(x)$ for the next round of data collection. This process is summarized in Algorithm 1.

We compare our strategy to i.i.d. sampling and two existing approaches commonly used in data curation. Both of these approaches propose a metric of example difficulty and prioritize difficult examples during data collection. The first metric, called Instruction-Following Difficulty (IFD) (Li et al., 2023), computes the ratio between the perplexity of training labels given inputs and the perplexity of only labels using a model finetuned for the task. The second metric uses heuristic notions of difficulty measured by external sources such as humans or more capable models (Chen et al., 2023; Lu et al., 2023; Zhao et al., 2023). For GSM8K, we use the number of lines in the target solution traces as a heuristic for difficulty, while for MATH, we use the difficulty levels provided in the dataset itself.

In Fig. 8, we evaluate the different data curation approaches for finetuning on GSM8k with Llama3 8B and MATH levels 1-3 with DeepSeekMath 7B (Shao et al., 2024). Our approach outperforms all three prior approaches, achieving $2\times$ the sample efficiency for the same target test accuracy

compared to i.i.d scaling in GSM8k, and $1.5\times$ sample efficiency on MATH levels 1-3. Furthermore, we find the gap in performances increases with dataset size, which suggests that better data curation metrics may become more important as models become more capable. These results highlight the effectiveness of pre-memorization accuracy as a criterion for targeted data collection, leading to enhanced generalization with fewer data points. We provide more details about our implementations in Appendix D.

## 6. Conclusion

Our work studies the relationship between learning dynamics and generalization in LLMs finetuned for reasoning tasks. We introduce the concept of pre-memorization train accuracy and show that it is a strong predictor of its test accuracy. We further show that a model's per-example pre-memorization train accuracy can be an indicator of the robustness model predictions for those examples. Finally, we leverage this insight for data curation, and show that prioritizing examples with low pre-memorization train accuracy can be more effective than i.i.d. data scaling and other data curation techniques. We hope that by providing a way attribute a model's generalization to specific aspects of the training process, our work can enable the design of more effective and principled training strategies.

## Impact Statement

This paper presents work whose goal is to advance the field of Machine Learning. There are many potential societal consequences of our work, none which we feel must be specifically highlighted here.

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

# A. Selection of Memorization Threshold

We find the threshold $p$ by sweeping across a range of values, calculating the pre-memorization train accuracy across different training runs, and selecting the value which yields the strongest predictor of test accuracy. In Fig. 9, we illustrate how the value of $p$ influences the $R^2$ for predicting average test accuracy. We can see that $R^2$ degrades smoothly with respect to $p$, which makes it is relatively easy to find a good value of p by sweeping a range of values.

This calibration process only requires a small number of training runs (e.g. 1-3) to arrive at a robust value of $p$ which can generalize to new training runs on the same model and finetuning dataset, illustrated in Fig. 10. However, it is important the the training runs used for calibration exhibit some spread over test accuracies, and memorization during training.

Finally, we also show that the calibration process generalizes to new test examples. We divide the test set into two halves: a calibration test set, and a heldout test set. We calibrate $p$ on the calibration test set, and evaluate the coefficient of determination of the heldout test set. In Fig. 11, we can see that the value of $p$ is able to generalize robustly to new examples on which it had not been calibrated, achieving high coefficient of determination.

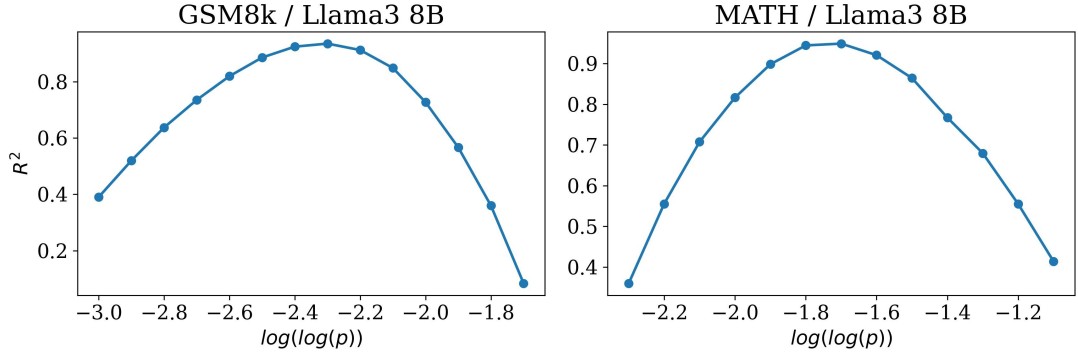

*Figure 9.* Relationship between the value of p and the coefficient of determination ($R^2$) with respect to pre-memorization train accuracy and test accuracy. The $R^2$ is taken in aggregate of all the corresponding training runs in Fig. 4.

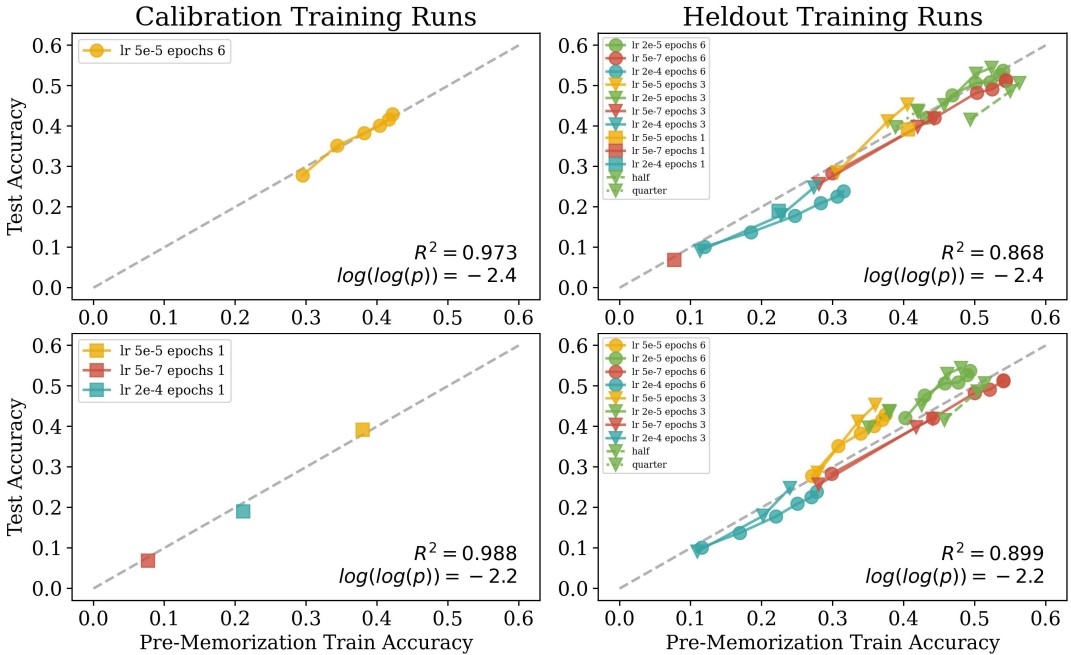

*Figure 10.* Calibrating $p$ on a subset of training runs, and evaluating $R^2$ on heldout training runs using GSM8k and Llama3 8B. We can see that calibrating on just 1-3 training runs was able to yield a robust value of $p$ which leads to high $R^2$ on heldout training runs.

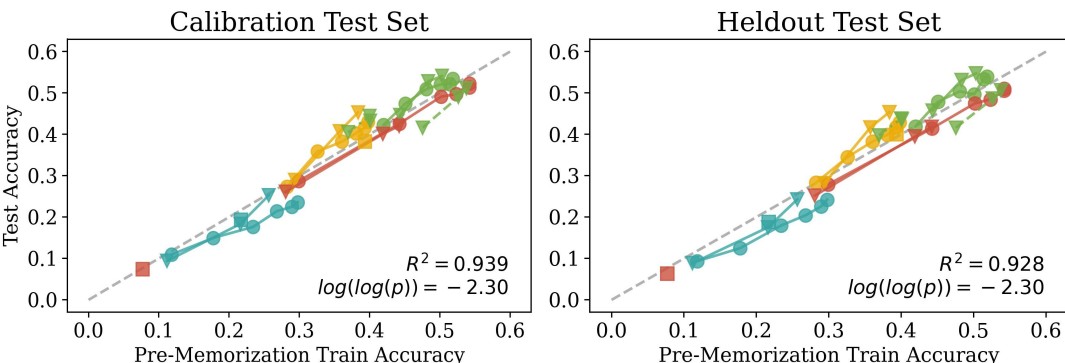

*Figure 11.* Calibrating $p$ using a subset of of the test set (calibration test set), and evaluating $R^2$ on a heldout test set using GSM8k and Llama3 8B. We can see that calibrating on just the calibration test set was able to yield a robust value of $p$ which leads to high $R^2$ on the heldout test set.

## B. Section 4.2 Training Runs Details

In this section, we will enumerate all training runs shown in Fig. 4 and their training details. For our half and quarter training runs, we fix the total number of training steps to be equivalent to training for 3 epochs on the full dataset.

### B.1. GSM8k LLama3 8B

For all training runs with GSM8k and Llama3 8B, we use the AdamW optimizer, with a linear decay learning rate scheduler with 20 warmup steps, a batch size of 128, and a max gradient norm of 2.

| Learning Rate | Epochs | Dataset Size |
|---------------|--------|--------------|
| 5e-5 | 6 | full |
| 2e-5 | 6 | full |
| 5e-7 | 6 | full |
| 2e-4 | 6 | full |
| 5e-5 | 3 | full |
| 2e-5 | 3 | full |
| 5e-7 | 3 | full |
| 2e-4 | 3 | full |
| 5e-5 | 1 | full |
| 5e-7 | 1 | full |
| 2e-4 | 1 | full |
| 2e-5 | 6 | half |
| 2e-5 | 12 | quarter |

### B.2. MATH LLama3 8B

For all training runs with MATH and Llama3 8B, we use the AdamW optimizer, with a linear decay learning rate scheduler with 20 warmup steps, a batch size of 24, and a max gradient norm of 2.

| Learning Rate | Epochs | Dataset Size |
|:---:|:---:|:---:|
| 5e-5 | 6 | full |
| 5e-7 | 6 | full |
| 2e-4 | 6 | full |
| 5e-5 | 3 | full |
| 5e-7 | 3 | full |
| 2e-4 | 3 | full |
| 5e-5 | 1 | full |
| 5e-7 | 1 | full |
| 2e-4 | 1 | full |
| 2e-5 | 6 | half |
| 2e-5 | 12 | quarter |

### B.3. GSM8k Gemma2 9B

For all training runs with GSM8k and Gemma2 9B, we use the Adam optimizer, with a cosine decay learning rate scheduler with (0.1*total steps) warmup steps, a batch size of 32, and a max gradient norm of 1.

| Learning Rate | Epochs | Dataset Size |
|:---:|:---:|:---:|
| 5e-4 | 6 | full |
| 5e-5 | 6 | full |
| 5e-6 | 6 | full |
| 5e-7 | 6 | full |
| 5e-4 | 3 | full |
| 5e-5 | 3 | full |
| 5e-6 | 3 | full |
| 5e-7 | 3 | full |
| 5e-4 | 1 | full |
| 5e-5 | 1 | full |
| 5e-6 | 1 | full |
| 5e-7 | 1 | full |
| 5e-5 | 6 | half |
| 5e-5 | 12 | quarter |

### B.4. MATH Gemma2 9B

For all training runs with MATH and Gemma2 9B, we use the Adam optimizer, with a cosine decay learning rate scheduler with (0.1*total steps) warmup steps, a batch size of 32, and a max gradient norm of 1.

| Learning Rate | Epochs | Dataset Size |
|:---:|:---:|:---:|
| 5e-4 | 6 | full |
| 5e-5 | 6 | full |
| 5e-6 | 6 | full |
| 5e-7 | 6 | full |
| 5e-4 | 3 | full |
| 5e-5 | 3 | full |
| 5e-6 | 3 | full |
| 5e-7 | 3 | full |
| 5e-4 | 1 | full |
| 5e-5 | 1 | full |
| 5e-6 | 1 | full |
| 5e-7 | 1 | full |
| 5e-5 | 6 | half |
| 5e-5 | 12 | quarter |

## C. Section 4.2 Prior Generalization Metrics

In this section we will more precisely describe each generalization metric.

### C.1. Gradient Variance

We calculate the gradient of the model for 5 different minibatches, take the variance across the 5 samples for each element of each weight matrix, and take the average over each element of the model weights.

### C.2. Distance from Initialization

We calculate the squared difference between each element of the model weights at initialization and after finetuning, and take the sun across all elements.

### C.3. Average Thresholded Confidence (ATC)

ATC computes a threshold on a score computed on model confidence such that the fraction of examples above the threshold matches the test accuracy. For the score, we use the likelihood of greedily sampled responses under the model. We calculate the the score over the training data using a model trained for 3 epochs using learning rate 2e-5, and calculate the threshold over the score using the test dataset. We then predict the test accuracies over different models in our experiment by calculating the score associated with the training data using each model, and measuring the percentage of examples whose score surpass the threshold that we previously calculated.

## D. Section 5.2 Implementation Details

For our approach for data curation, we implemented the process described in Algorithm 1, with 5 iterations ($n$) and using threshold ($t$) 0.75 for both GSM8k and MATH.

For the IFD approach for data curation, we calculated the IFD score using a model that was train on the test set associated each dataset for 2 epochs. This is because, in order to calculated the IFD score, we need a model which has been briefly trained for the task of interest, but which has not been exposed to the dataset for which we want to calculate the IFD score over. Note that this model is only used for calculating for the IFD score, and not used for evaluations in our experiments, so there is no data leakage.

For both the IFD approach and the heuristic approach, we take $P'(x)$ to be top 50 percentile of examples for GSM8k, and top 75 percentile of examples for MATH. We designed these percentiles to roughly match the percentile of examples that our approach selects from.

For all training runs, we use the AdamW optimizer, with a linear decay learning rate scheduler with 20 warmup steps, a batch size of 128, a max gradient norm of 2, a learning rate of 2e-5, and 3 epochs of training.

