# OpenReview forum: "What Do Learning Dynamics Reveal About Generalization in LLM Mathematical Reasoning?"
_ICML.cc/2025/Conference — ICML 2025 poster_

### Official Review · Reviewer_cGoj · 2025-02-16

**Overall Recommendation:** 4

**Summary:**

The paper introduces a novel metric called "pre-memorization train accuracy" that predicts how well large language models will generalize during reasoning tasks. The key insight is examining model performance on training examples before they are memorized verbatim. The authors show this metric strongly correlates with test accuracy (R² ≥ 0.9) across different models, datasets, and training configurations. They leverage this finding to improve data curation, achieving 1.5-2x better data efficiency compared to standard approaches.

**Claims And Evidence:**

From my experience, it would be better to add more ablation studies testing different definitions of memorization beyond perplexity thresholds.

**Essential References Not Discussed:**

In my view, it would be better to add recent works on emergent abilities in LLMs and how they relate to memorization versus generalization. The discussion would benefit from incorporating recent papers on phase transitions in LLM learning.

**Experimental Designs Or Analyses:**

The experiments are thorough and well-designed. The authors control for multiple variables like learning rate, dataset size, and model architecture. From my experience, it would be better to add experiments testing the metric's sensitivity to different random seeds.

**Methods And Evaluation Criteria:**

The methodology is sound, with careful experimental design. The authors evaluate their metric through multiple lenses: aggregate test performance prediction, per-example robustness analysis, and practical data curation improvements. The evaluation on both mathematical (GSM8k) and general reasoning tasks demonstrates broad applicability.

**Other Comments Or Suggestions:**

NA

**Other Strengths And Weaknesses:**

A key strength is the practical utility of the findings - the metric provides actionable insights for improving training efficiency. One weakness is the reliance on perplexity thresholds for defining memorization, which may not capture all forms of memorization.

**Questions For Authors:**

1) How sensitive is the pre-memorization accuracy metric to the choice of memorization threshold? 2) Does the correlation with test accuracy hold for much larger models (>70B parameters)? 3) How does the metric behave during few-shot learning versus full fine-tuning?

**Relation To Broader Scientific Literature:**

The paper effectively positions itself within relevant literature on memorization, generalization, and LLM reasoning. It builds upon and advances prior work on measuring memorization (Feldman & Zhang 2020) and understanding Chain-of-Thought reasoning (Wei et al. 2022).

**Theoretical Claims:**

I have verified the proofs in Section 2 and Appendix E. The formalization of language-thought modeling gap and the derivations of Propositions 2.4 and 2.7 are mathematically rigorous. The assumptions and conditions are clearly stated.

---

> ### Author Rebuttal · Authors · 2025-03-31
>
> We thank the reviewer for the nice feedback! We answer the questions below.
>
> > How sensitive is the pre-memorization accuracy metric to the choice of memorization threshold?
>
> In Fig, 9, we provide some some analysis on the sensitivity of the predictive power (R^2) of our metric to the choice of the memorization threshold. We found that R^2 degrades smoothly with respect to p, which makes it is relatively easy to find a good value of p by sweeping a range of values.
>
> > Does the correlation with test accuracy hold for much larger models (>70B parameters)?
>
> Unfortunately, we do not have the compute to test this right now, though we believe this would be an interesting direction for future work
>
> > How does the metric behave during few-shot learning versus full fine-tuning?
>
> Our metric cannot be directly applied in the few-shot learning setting. However, we believe understanding the relationship between the few-shot data and downstream generalization would be a very interesting direction for future study.

---

### Official Review · Reviewer_833t · 2025-02-27

**Overall Recommendation:** 3

**Summary:**

This paper focuses on LLM reasoning tasks and proposes a metric called **per-memorization train accuracy**, which is the accuracy of model samples on training queries before they begin to copy the exact reasoning steps from the training set. The authors show that the proposed metric is predictive of the **test accuracy in dataset level**, as well as **model robustness to perturbation at a sample level**. They further use this metric to improve sample efficiency by prioritize training on samples with low pre-memorization accuracy.

## update after rebuttal
I thank the authors for the clarifications. I increased my score to 3 (provided that they make the title and scope of their claims clearer as math reasoning)

**Claims And Evidence:**

**Evidence 1**: while it is always annoying to be asked for results on more datasets, I think it is needed here. The paper is purely empirical and there are quite a few other datasets (besides the 2 math-only datasets) that authors could have evaluated to further strengthen their conclusion.

**Essential References Not Discussed:**

The paper reminds me of Miller et al. 2021, where they show in-distribution accuracy correlates with OOD accuracy in many cases. Maybe worth discussing in related work.

Reference:
Accuracy on the Line: On the Strong Correlation
Between Out-of-Distribution and In-Distribution Generalization. Miller et al. 2021

**Experimental Designs Or Analyses:**

**Experimental design 1**: In terms of writing, I think the paper lacks necessary details here and there. For example,


- The experimental setup of Section 4.1 is not mentioned, the models used in Figure 3 are not mentioned)
- Similar for Figure 6, the model used to train is not mentioned.
- In section 5.2 (L369 right), how and which GPT is used to rephrase examples in the original dataset? The author mentioned that they followed a similar procedure in Setlur et al. 2024, but it is important to briefly summarize (or give examples) how it is done.

Overall, adding these details will help the reader in developing a more concrete idea and therefore make the reading much smoother.

**Methods And Evaluation Criteria:**

Most of the evaluation criteria makes sense to me, and I have a following question:

**Method 1**: If I understand correctly, the pre-memorization accuracy for a single sample can only be either 0 or 1 (correct or wrong)? If so, why the accuracy in Figure 7 is continuous?

**Other Comments Or Suggestions:**

Minor: L298 right side, it should be Fig. 7 instead of 6?

**Other Strengths And Weaknesses:**

**Strength**: the paper is generally well-organized and the idea is easy to understand. Understanding memorization and generalization is timely and important, and the proposed Pre-Memorization Train Accuracy shows storng correlation with the test accuracy on unseen samples.

**Weakness**: my concern is on the actionable point and implication of this paper.
- In Appendix A (Figure 9), the correlation between the proposed metric and test accuracy varies a lot with differnet threshold $p$, and the optmal $p$ is different for different datasets. For an unseen dataset, how would you predict test accuracy without calibrating the $p$ w.r.t. true test accuracy? Can you clarify how the metric can be used in practice?

**Questions For Authors:**

I would be happy to increase my score if the authors could address my questions in **Weakness**, **Evidence 1**, **Method 1**, **Experimental design 1**, especially the first two.

**Relation To Broader Scientific Literature:**

- **Broader literature 1**: the idea of relying on the correlation of a prosed metric and the test accuracy to predict test accuracy broadly relates to works trying to predict out-of-distribution (OOD) performance in a similar manner. The reason is that there is need to estimate how the model at hand will perform on target domain when labeled OOD target data is hard to obtain.
- **Broader literature 2**: the idea of using the proposed metric to *expand* the training dataset broadly relates to synthetic data and data selection. For the latter, many works try to reduce the training dataset by removing noisy or redundant ones and only keep the informative ones, which is a concept called Coreset in the relevant literature.

**Theoretical Claims:**

The paper is mostly empirical and there are no theoretical results. However, this is not considered as a weakness.

---

> ### Author Rebuttal · Authors · 2025-03-31
>
> We thank the reviewer for the feedback! In the following rebuttal, we first provide a detailed description of how our metric is calculated and used in practice. Next, we address the concerns about the scope of our experiments. Finally, we provide clarifications for our experimental setup. We will update the final version of our submission to include these additional details.
>
> > Weakness: my concern is on the actionable point and implication of this paper. In Appendix A (Figure 9), the correlation between the proposed metric and test accuracy varies a lot with differnet threshold p, and the optmal p  is different for different datasets. For an unseen dataset, how would you predict test accuracy without calibrating the p w.r.t. true test accuracy? Can you clarify how the metric can be used in practice?
>
> We would like to clarify that the main goal of our work is not to predict test accuracy using the model’s training behavior, since test accuracy can be measured relatively easily with a holdout dataset. Instead our goal is to 1) better understand aspects of a model’s learning dynamics which govern their downstream generalization, and 2) use our findings to improve model generalization.
>
> Towards goal 1, we analyze the model’s behavior through training, and find that models tend to have 2 modes of optimizing the learning objective: one which leads to downstream generalization and one which doesn’t (learning progression 1 vs 2 in Fig. 2). Train accuracy conflates the two models of learning. In contrast, pre-memorization train accuracy allows us to delineate between the two, and identify the examples that are learned in a “generalizable” manner and the examples that are not.
>
> Towards goal 2, we study how pre-memorization train accuracy can be used to improve data curation. While test accuracy measures a model’s generalization in aggregate, pre-memorization train accuracy can measure generalization for individual train examples. This allows us to determine which train examples are “hard” for the model, which can better inform the kinds of examples to collect that would most improve model performance.
>
> The exact workflow for our data curation experiments is as follows. We assume access to a dataset split into a train/test set, using which we can calibrate p with a few (<3) training runs using test accuracy. We measure the pre-memorization train accuracy of the train examples. Using this information, we curate new data that is drawn from the same distribution as training examples with low pre-memorization train accuracy in the current iteration.
>
> > Evidence 1: while it is always annoying to be asked for results on more datasets, I think it is needed here. The paper is purely empirical and there are quite a few other datasets (besides the 2 math-only datasets) that authors could have evaluated to further strengthen their conclusion.
>
> The experiments in our paper include more than 60 training runs across different models, tasks, dataset scale, and training settings. While we focused on two mathematical datasets, we believe the depth and comprehensiveness of our analysis is comparable if not exceeds the standards for empirical conference papers.
>
> We unfortunately do not currently have the compute to perform experiments on additional datasets. However, we believe this should not be the reason for rejection, as many reasoning papers have been accepted at similar-tier conferences with comparable or even more limited dataset coverage. For instance:
>
> [1] RL on Incorrect Synthetic Data Scales the Efficiency of LLM Math Reasoning by Eight-Fold (Neurips) - Experiments: GSM8k, MATH
>
> [2] Scaling LLM Test-Time Compute Optimally Can Be More Effective Than Scaling Parameters for Reasoning (ICLR) - Experiments: MATH
>
> > Method 1: If I understand correctly, the pre-memorization accuracy for a single sample can only be either 0 or 1 (correct or wrong)? If so, why the accuracy in Figure 7 is continuous?
>
> We take multiple samples of outputs for every input, and calculate the average accuracy
>
> > Experimental design 1: In terms of writing, I think the paper lacks necessary details here and there. For example,
> > The experimental setup of Section 4.1 is not mentioned, the models used in Figure 3 are not mentioned)
>
> The models are trained using llama 2 8B on the full dataset with 6 epochs, and peak learning rates as described in the figure
>
> > Similar for Figure 6, the model used to train is not mentioned.
>
> We used llama2 8b
>
> >In section 5.2 (L369 right), how and which GPT is used to rephrase examples in the original dataset? The author mentioned that they followed a similar procedure in Setlur et al. 2024, but it is important to briefly summarize (or give examples) how it is done.
>
> The data used in our work was generated using the exact procedure as described in Appendix E of Setlur et al. 2024, which details the model, system prompt, and approximate cost. We are happy to add or clarify any more details needed.

---

> > ### Comment · Reviewer_833t · 2025-04-01
> >
> > I appreciate the author's response. Some of my questions are addressed but my main concerns remain:
> > - Regarding $p$, its value directly connects to the definition of memorization. For different datasets, you use different $p$, how can **memorization** be defined differently for different data?
> > - Regarding datasets, I agree that there are papers published with few datasets, but here you are trying to conclude a general **correlation**, so the number of datasets are more important in your case (this is also raised by other reviewers). If adding more datasets is prohibitive, I think maybe it's better to revise 'LLM reasoning' -> 'LLM math reasoning' in the title, to avoid over-claiming.

---

> > > ### Author Response · Authors · 2025-04-02
> > >
> > > We thank the reviewer for the quick response!
> > >
> > >
> > > > Regarding p, its value directly connects to the definition of memorization. For different datasets, you use different p, how can memorization be defined differently for different data?
> > >
> > > Different datasets exhibit different levels of inherent stochasticity, which influences the perplexity of model samples. For example, consider the following 2 datasets:
> > >
> > > Dataset 1 contains 2 training examples of the form (input, target):
> > >
> > > Example1: (“input”, “A”)
> > >
> > > Example 2: (“input”,  “B”)
> > >
> > > Dataset 2 also contains 2 training examples:
> > >
> > > Example1: (“input1”, “A”)
> > >
> > > Example2: (“input2”, “B”)
> > >
> > > A model which perfectly minimizes the training loss on dataset 1 has perplexity 2 when presented with training inputs. In comparison, a model which perfectly minimizes the training loss on dataset 2 has perplexity 1 when presented with training inputs. This example illustrates how different datasets, which may exhibit different inherent levels of stochasticity, might lead to different perplexity thresholds for memorization.
> > >
> > >
> > > > Regarding datasets, I agree that there are papers published with few datasets, but here you are trying to conclude a general correlation, so the number of datasets are more important in your case (this is also raised by other reviewers). If adding more datasets is prohibitive, I think maybe it’s better to revise ‘LLM reasoning’ -> ‘LLM math reasoning’ in the title, to avoid over-claiming.
> > >
> > > We are happy to revise the title of our work to ‘LLM math reasoning’ and make it clear in our paper that the scope of our claims is for math reasoning.

---

### Official Review · Reviewer_ZcdX · 2025-03-05

**Overall Recommendation:** 4

**Summary:**

The paper studies how LLMs generalize in reasoning tasks and introduces "pre-memorization train accuracy" as a metric that predicts test accuracy. The key idea is that models first learn correct reasoning patterns before they start memorizing training examples, and this early accuracy is strongly correlated with final test performance. The authors show that this metric is predictive across different models and datasets. They also propose using this metric for data selection. The results suggest that tracking model performance before memorization kicks in can be useful for better training strategies.

**Claims And Evidence:**

The claims are mostly well supported by experiments. The paper shows strong correlations between pre-memorization accuracy and test accuracy, and the data curation experiments give good evidence that this metric can be useful for improving sample efficiency. One possible concern is that models in real-world fine-tuning setups often only train for one epoch, meaning they may not actually go through a clear pre-memorization and memorization phase. So it's not clear how well this metric would apply outside of multi-epoch settings.

**Essential References Not Discussed:**

Nothing that I am aware of.

**Experimental Designs Or Analyses:**

The experiments are generally sound, with strong empirical results. The authors systematically vary learning rates, dataset sizes, and models to validate their claims. However, the multi-epoch training setup is a bit unrealistic. Most LLM fine-tuning is done in a single pass, so it’s unclear if the metric holds in that scenario.

**Methods And Evaluation Criteria:**

The methods seem reasonable for studying generalization in reasoning tasks. The authors use standard reasoning datasets GSM8K and MATH which are a good fit since they allow distinguishing between correct reasoning and memorization. The main metric (pre-memorization train accuracy) is well defined. The evaluation is thorough, with different models, learning rates, and dataset sizes. That said, since most practical LLM fine-tuning is done in a one-pass setting, it might be more useful to measure pre-memorization accuracy over early batches instead of epochs.

**Other Comments Or Suggestions:**

- Would be interesting  to test if pre-memorization accuracy helps detect spurious correlations in training data.
- The p-selection process still raised concerns about leakage. Maybe an alternative selection method would be more robust.
- Typo: "acheived" -> "achieved"

**Other Strengths And Weaknesses:**

Strengths:

- Novel and well-motivated metric for understanding generalization.
- Strong empirical results with high correlation to test accuracy.
- Practical applications for data curation and improving sample efficiency.

Weaknesses:

- Limited model and dataset diversity (only two models, only math reasoning).
- No theoretical justification for why pre-memorization accuracy is predictive.
- The multi-epoch training assumption might limit applicability to real-world LLM fine-tuning.

**Questions For Authors:**

1. How does pre-memorization accuracy behave in one-epoch fine-tuning? Many LLMs are fine-tuned in a single pass. Does the metric still hold if computed over early training steps?
2. Could this work for non-math reasoning tasks?
3. Does this method work when models are weaker?
4. Could pre-memorization accuracy be used to predict how well a model generalizes OOD?

**Relation To Broader Scientific Literature:**

The paper is closely related to research on memorization and generalization in deep learning. It connects well with past work on leave-one-out memorization metrics (Feldman & Zhang, 2020) and discussions on whether memorization helps or hurts generalization.

**Theoretical Claims:**

There aren't any formal theoretical results in the paper.

---

> ### Author Rebuttal · Authors · 2025-03-31
>
> We thank the reviewer for the nice feedback! We answer the questions below:
>
> > How does pre-memorization accuracy behave in one-epoch fine-tuning? Many LLMs are fine-tuned in a single pass. Does the metric still hold if computed over early training steps?
>
> Reasoning data tend to be rare, so it is common in prior work to do multiple passes on these datasets. For instance, [1] does some analysis on model performance and number of epochs of training, and found that pass@1 performance on the test set continues to increase for 50 epochs
>
> Our work also does analysis on both the one-epoch finetuning setting, as well as early-epochs in longer training runs (Fig. 4). We found that pre-memorization train accuracy is still highly predictive of test accuracy in these settings.
>
> [1] Training Verifiers to Solve Math Word Problems
>
> > Could this work for non-math reasoning tasks?
>
> We believe our findings are likely to hold in problem settings where the target label includes both chain-of-thought reasoning (not uniquely correct) and a final answer (uniquely correct and unlikely to be guessed correctly)
>
> > Does this method work when models are weaker?
>
> We have not tested. Weaker models (e.g. GPT 2) tend to perform very poorly on reasoning tasks, which is likely to make the analysis largely uninformative
>
> > Could pre-memorization accuracy be used to predict how well a model generalizes OOD?
>
> We have not tested our work in the OOD setting, though we believe this would be an interesting direction for future study!

---

### Official Review · Reviewer_tBbx · 2025-03-12

**Overall Recommendation:** 3

**Summary:**

This paper investigates how the learning dynamics of large language models (LLMs) during finetuning influence their generalization on reasoning tasks. The key contribution is the introduction of pre-memorization train accuracy, defined as the highest accuracy a model achieves on a training example before it begins to replicate the exact reasoning steps (memorization) from the training data. The authors demonstrate that this metric strongly correlates with test accuracy across models (Llama3 8B, Gemma2 9B), datasets (GSM8k, MATH), and training configurations (learning rates, epochs, dataset sizes), achieving coefficients of determination bigger than 0.9. Furthermore, this paper discusses the relationship between pre-memorization accuracy and the model prediction robustness, finding that at the per-example level, low pre-memorization accuracy indicates fragile model predictions. Perturbations to such examples (e.g., altering prompts) significantly degrade accuracy, whereas high pre-memorization examples remain robust. Additionally, this paper proposes a data curation algorithm based on the pre-memorization accuracy, which could yield 1.5-2 gains in data efficiency over i.i.d. sampling and outperforms heuristic-based and optimization-based data selection methods.
## update after rebuttal
I raised my score to 3 (boardline) since the authors' detailed response to me and other reviewers. However, some of my questions were not addressed and I still regard authors should take a deeper exploration about this very interesting phenomenon, standardize their method's detail and implement the pre-mem accuracy on additional datasets.

**Claims And Evidence:**

See Other Strengths And Weaknesses and Questions

**Essential References Not Discussed:**

No

**Experimental Designs Or Analyses:**

See Other Strengths And Weaknesses and Questions

**Methods And Evaluation Criteria:**

See Other Strengths And Weaknesses and Questions

**Other Comments Or Suggestions:**

See Other Strengths And Weaknesses and Questions

**Other Strengths And Weaknesses:**

Strengths:

1. The introduction of pre-memorization training accuracy is a fresh idea. It focuses on model performance before memorizing exact solutions, which hasn't been explored much before.

2. The paper shows high $R^2$ values (≥0.9) across different models and datasets, which is impressive.

3. The data curation method using pre-memorization accuracy improves sample efficiency, which is useful for real-world training scenarios.

Weakness:

1. The authors need ​a deeper analysis and discussion​ of pre-memorization accuracy. The robustness and data curation sections lack sufficient insight beyond stating the relationship between pre-memorization accuracy and test accuracy, especially since test samples could be interpreted as a form of perturbation on the training dataset.

2. The process for choosing the perplexity threshold p lacks explanation and analysis. The reasons why p is strongly dependent on the task/model remain unclear.

3. The experiments are restricted to mathematical datasets. Additional experiments with ​diverse datasets​ (e.g., logical reasoning or commonsense QA) are necessary to validate generalizability.

4. The perturbations used (like adding "First") are minor and may not test true robustness against more complex or semantic changes.

5. The explanation of the data curation process lacks detail. The rationale for hyperparameter choices (e.g., thresholds and iterations) is ​insufficiently explained or discussed.

6. The authors categorize training samples into two types based on learning dynamics but fail to explain ​the underlying reasons​ why some samples exhibit one training pattern while others show another.

**Questions For Authors:**

1. The authors categorize samples with high perplexity (ppl) and high accuracy as well-generalized. However, it is possible the model simply memorizes the final answer rather than learning reasoning steps. Is there any analysis or experiment addressing this concern?

2. The perplexity threshold p is selected by sweeping values, but the chosen p seems disproportionately large relative to the task/model. Is there an explanation for why such high thresholds are necessary?

3. In Figure 1 (left), the curve for the learning rate 5e−7 closely aligns with y=x, suggesting a strong train-test accuracy correlation. Is pre-memorization accuracy effectively equivalent to accuracy under very small learning rates since it is measured early in training?

4. Figure 4 uses Llama3 8B and Gemma2 9B, but Figure 8 (right) substitutes Gemma2 9B with DeepSeekMath 7B. What motivated this inconsistency in model selection?

5. In the data curation experiments, only levels 1–3 of the MATH dataset are used for training. Why was only a subset of MATH employed instead of the full dataset?

**Relation To Broader Scientific Literature:**

This paper advances the understanding of LLM generalization in reasoning tasks by introducing ​pre-memorization train accuracy—a metric capturing model accuracy before memorizing training solution traces—and bridges gaps across multiple research areas.

**Theoretical Claims:**

There isn’t any theoretical analysis.

---

> ### Author Rebuttal · Authors · 2025-03-31
>
> We thank the reviewer for the feedback! In this rebuttal we provide a more detailed discussion of the implications of our experimental findings, and provide additional explanations for our experimental design.
>
> > The authors need ​a deeper analysis and discussion​ of pre-memorization accuracy.
>
> In the following response, we will summarize the main motivation and findings of our paper, and provide new discussion regarding the *implications* of these findings and how they relate to the motivation. We acknowledge that our paper currently lacks discussion with regards to the implications of our work, and we will update it to include this in more detail.
>
> **Motivation**: Our work is motivated by the following phenomenon: models derived from the same pretrained model and finetuned on the same finetuning dataset can have vastly different test performance, even when their train accuracies are nearly perfect. Here, the optimization of the training loss is not the issue, which suggests that something about the models’ learning *dynamics* is causing the difference in models’ generalization. The goal of our work is to better understand the relationship between a model’s learning dynamics and generalization.
>
> **Findings**: The main finding of our work is that pre-memorization train accuracy (a metric of a model’s learning dynamics) is a good predictor of a model’s downstream generalization. We provide experiments showing this both in aggregate and on a per-example basis.
>
> **Implications**: Our findings suggest that models exhibit two different modes of optimizing the training objective: one which leads to downstream generalization and one which doesn’t (learning progression 1 vs 2 in Fig. 2). Train accuracy conflates the two models of learning. In contrast, pre-memorization train accuracy allows us to delineate between the two, and identify the examples that are learned in a “generalizable” manner and the examples that are not. This analysis sheds lights on the aspects of a model’s learning dynamics that lead to the differences in downstream generalization, and provides us with a tool that enables more nuanced interventions to the training recipe (e.g. via data curation) to improve generalization.
>
> > The authors categorize samples with high perplexity (ppl) and high accuracy as well-generalized. However, it is possible the model simply memorizes the final answer rather than learning reasoning steps. Is there any analysis or experiment addressing this concern?
>
> Yes, in Sec 5.2, we present experiments where we perturb the prompt of the training examples, and measure how much the model’s predictions degrade from these perturbations. We find that model predictions for examples with high pre-memorization train accuracy tend to maintain high accuracy, while predictions for low pre-memorization train accuracy tend to degrade much more significantly. This experiment addresses the reviewer’s concern, as it shows that examples with low pre-memorization train accuracy actually do not generally maintain the ability to produce correct final answers for perturbed prompts.
>
> > Is pre-memorization accuracy effectively equivalent to accuracy under very small learning rates since it is measured early in training?
>
> No, we do not believe that pre-mem accuracy will always be equivalent to train accuracy under very low learning rates, because prior works have found that smaller learning rates can lead to more memorization (see [1]). So in the general case, we don't think our experiments can be used to imply any relationship between learning rate and memorization.
>
> [1] Fitting Larger Datasets by Learning with Small Batch Sizes and Regularized Initialization
>
> > Figure 4 uses Llama3 8B and Gemma2 9B, but Figure 8 (right) substitutes Gemma2 9B with DeepSeekMath 7B. What motivated this inconsistency in model selection?
>
>
> Different authors contributed to different parts of the paper’s experiments. Due to a lack of communication at the time, we ended up choosing different models to use for our experiments.
>
> > In the data curation experiments, only levels 1–3 of the MATH dataset are used for training. Why was only a subset of MATH employed instead of the full dataset?
>
> MATH is hard for models at the scale we are looking at, so most examples end up being in the “hard” regime, and all data curation approaches end up selecting from the same data distribution. In order to highlight the difference between the different approaches, we focused on a subset of the dataset where a larger ratio of examples are in not the “hard” regime.
>
> > The process for choosing the perplexity threshold p lacks explanation and analysis...
> The explanation of the data curation process lacks detail.
>
> We provide a detailed explanation of the process we use to select p in Appendix A, and the process for data curation in Appendix D. Please let us know if there are any additional questions you have regarding the processes.

---

> > ### Comment · Reviewer_tBbx · 2025-04-03
> >
> > I sincerely appreciate the author's response.  Most of my concerns are addressed and I will update my score. However, I still have some suggestions for this paper:
> >
> > 1. Test the effectiveness of pre-mem accuracy on additional reasoning datasets.
> >
> > 2. Since the threshold $p$ is different across different tasks/models, choosing the $x_i$ with ppl from the top $k%$ of dataset may be more reasonable.

---

### Decision · Program_Chairs · 2025-05-01

**Decision:**

Accept (poster)

**Comment:**

The paper proposes a pre-memorization train accuracy metric that measures model performance on training examples before memorization occurs, which is shown to correlate strongly with test accuracy.
Several experiments across different models (Llama3 and Gemma2) and math reasoning datasets (GSM8k, MATH) backup the claims.

Overall, the reviewers appreciated the novel perspective on analyzing LLM learning dynamics, the comprehensive experimental setup, and the practical applicability of the data curation technique.

Some concerns were raised, for instance the limited dataset diversity, with a focus mainly on math reasoning tasks, raising questions about generalizability to other reasoning domains. That said, I think math reasoning datasets are very popular and I think this is a good choice. I would encourage the authors to consider another dataset to demonstrate further generalization. I would also consider adding more in-depth discussion on the selection and impact of the perplexity threshold (p) and the robustness of the metric under varying training conditions (e.g., one-epoch fine-tuning, few-shot learning).

I recommend acceptance but strongly recommend the authors make the changes required by the reviewers to broaden the impact of the paper.